# The Influence of Risk Culture on the Performance of International Joint-Venture Securities

**Xiaoteng Ma [1], Ziyu Tang [2], Dan Wang [3],\* and Hao Gao [4]**

1   School of Economics and Management, Tsinghua University, Beijing 100084, China;
    maxt.16@sem.tsinghua.edu.cn
2   College of Social Sciences, Adam Smith Business School, University of Glasgow, Glasgow G37ld, UK;
    Ziyu.Tang@glasgow.ac.uk
3   International Business School, University of International Business and Economics, Beijing 100029, China
4   PBC School of Finance, Tsinghua University, Beijing 100083, China; gaoh@pbcsf.tsinghua.edu.cn
\*   Correspondence: wangdan@uibe.edu.cn; Tel.: +86-010-6449-4330

**Abstract:** With the development of economic globalization, culture is a key factor supporting the sustainability of foreign direct investment (FDI), especially for multinational enterprises. This paper takes the Chinese capital market as a sample and, combined with interviews with managers of international joint-venture securities (IJVS), finds that the culture of participants formed in developed and emerging capital market has a significant impact on the performance of IJVS. Using the degree of price fluctuation to measure the risk culture of each capital market, this paper observes that the risk culture in the Chinese capital market is significantly stronger than that of developed countries. This paper also finds that the stronger the risk culture IJVS shareholders have, the better they can adapt to the environment of the Chinese capital market and the better the performance they can achieve. Furthermore, risk culture distance, calculated by the risk culture differences between foreign shareholders and Chinese capital market, are significantly negatively correlated with IJVS performance and efficiency.

**Keywords:** multinational enterprises; business performance; technical efficiency; risk culture

## 1. Introduction

The term "sustainable development" originated in Germany; its original purpose was to protect the forest while simultaneously being successfully implemented in the field of cross-border cooperation and culture as well [1]. Determining the factors that could affect the performance and efficiency of cross-border cooperation is a valuable research topic. Indeed, scholars have discovered many hard factors that can affect cross-border cooperation, for instance, border areas, organizational structure, cross-border procedures, and related supporting mechanisms [2,3], while some soft factors also play a significant role. As a type of cross-border cooperation by FDI, IJVS regularly faces cross-border management involvement [4]. This makes the performance even more susceptible to various factors, for example, culture [5–9]. Although many scholars found the impact of culture on performance, the results remain mixed, especially for emerging markets, and to the best of our knowledge, there are no studies investigating the role of cultural difference in explaining the poor performance of IJVS. This paper intends to demonstrate how culture acts as a key variable supporting sustainable cross-border cooperation, especially for FDI of IJVS.

Due to strict regulations in China, foreign financial institutions must join the Chinese capital market as IJVS. Ideally, these established IJVS provide not only opportunities for foreign financial institutions to explore the Chinese market, but also advanced management experience in the capital

market. However, the performance of these IJVS are not well, and a number of IJVS have faced years of losses. A recent study shows that the corporate culture of different financial institutions varies greatly and has a direct impact on service quality and performance. The cultural difference of securities companies should be especially great among institutions from different countries that share an intrinsic cultural gap, which could therefore cause culture conflict among IJVS with institutional shareholders from different cultural backgrounds [10].

At the beginning, we interviewed managers from different IJVS and found that there is great divergence between Chinese and foreign shareholders on the judgment of risks. It is common to see foreign shareholders abandon businesses because of risk compliance issues, even if those businesses are widely accepted by other Chinese competitors. Additionally, they lose some opportunities due to the lengthy risk compliance process. Managers attributed this result to cultural differences.

Culture is a psychological process shared by people with the same educational background or experience, which determines individuals' decision-making style and behavior [11]. Consistent with predictions from the psychology literature, culture can actually influence the co-movement of stock prices and the information environment of the stock market by affecting the correlations among investors' trading activities. For example, individualism is associated with higher firm-specific variation [12]. Hofstede [13] employed four dimensions to demonstrate national culture: Power distance, uncertainty avoidance, individualism versus collectivism, and masculinity versus femininity. The characteristics of uncertainty avoidance are the most relevant to capital markets since the intrinsic nature of risk is uncertainty [14–16]. Therefore, based on the meaning of culture and the impact mechanism of culture on financial markets, this paper defines the psychological process that participants rely on to judge uncertainty under the specific capital market environment as "risk culture." We believe this risk culture has a direct impact on participants' risk preference and decision- making.

The recent establishment of China's financial market was rather late, for example, the Shanghai Stock Exchange was established in 1990, and we believe Chinese participants have stronger risk preference and tolerance because of the poor institutional quality—that is, higher policy uncertainty [17] and an imperfect legal system. In particular, the incomplete formal system provides the informal system with greater space and further increases market uncertainty; therefore, the risk culture in the Chinese capital market is relatively strong.

Shareholders of IJVS have different risk cultures, which are determined by their home capital market. Therefore, those shareholders face "multilayered acculturation"; i.e., they embody the culture of both the host country and their home country [18]. As China's financial market has a strong risk culture, we believe that IJVS are likely to gain advantages when foreign shareholders' risk culture is similar to that of China.

Furthermore, although foreign shareholders may not have controlling shares of IJVS, they have actual decision-making rights. However, the selection of projects is usually the result of compromise between Chinese and foreign shareholders. IJVS usually adopt joint review and collective decision-making in management, which is quite different from the Chinese shareholders' business philosophy of highlighting the main person in charge; that is, the power distance of culture is different [13]. Moreover, we found that foreign and Chinese shareholders often have different value judgments of the same information; for example, during the interviews, we found that their attitudes toward city development bond, a bond with Chinese characteristics, are different. This cultural difference caused IJVS to lose some valuable projects, and the consequences grow as the cultural differences between the two sides' shareholders widen.

The data we used to test the above hypothesis, published by the China Securities Association, cover 2006 to 2016 and contains 10 IJVS. Their operating performance was measured by return on assets (ROA) [19,20], while their technical efficiency (TE) was measured by the stochastic frontier analysis (SFA) model [21,22]. We measured the participants' risk preference in the different capital markets, which we define as risk culture, by calculating the degree of price fluctuation brought by recessive factors [23,24].

Our preliminary conclusion shows that compared with developed countries, the Chinese capital market has a stronger risk culture. We further approximated the shareholders' risk culture by their home countries' capital markets. The regression results show that in the Chinese financial market, when the foreign shareholders' risk culture is stronger, the ROA and technical efficiency of IJVS are higher. Finally, we found that the difference in risk culture between IJVS' foreign shareholders and the Chinese capital market has a significant influence on firm performance. The smaller the difference, the higher the firm's performance and technical efficiency. Besides, this paper further decomposed the performance of securities companies from the time and business dimensions. We found that IJVS have significantly lower technical efficiency and performance compared to Chinese securities companies. We also found that their businesses are less diversified, and even in their focused business area, their performance is still significantly lower. All of this analysis proves that a soft factor, risk culture, affects the efficiency and performance of IJVS. Additional testing shows that this negative impact is aggravated when IJVS promote their business pluralism.

To solve the self-selection bias, we matched IJVS with 20 comparable Chinese securities according to the date of establishment, place of headquarters, and asset scale. New analysis also shows that the shareholder risk culture has a significantly positive influence on the company's ROA and technical efficiency. This paper used geographical distance as an instrumental variable to alleviate the endogeneity problem. The results also indicate that cultural distance dampens the performance of IJVS.

Our research makes the following contributions. First, it fills a deficiency in existing studies of the performance of multinational enterprises in emerging markets [25] and offers a new explanation for the factors affecting the performance of financial institutions. Second, it provides a new perspective for research on cultural impact [26] and enriches the literature in this field. Third, the conclusions of this paper could help to encourage multinational corporations to emphasize the role of culture in their business, helping more companies make rational prejudgments of their capacities in host countries, and therefore promote sustainable FDI.

The remainder of the paper is organized as follows. Section 2 introduces the literature review, and develops the hypotheses. Section 3 describes the methodology, data, and variables. Section 4 presents the empirical results, and Section 5 concludes the paper.

## 2. Literature Review and Hypotheses

Culture is a "collective mental programming" shared by many people who have the same education and life experience [11]. People with a specific culture will form a fixed pattern of principles and values, internalized via thinking and feeling, and externalized through practices that make them different from other groups or organizations [27]. Guiso et al. [28] offered a similar definition of culture: A set of customary beliefs formed in a racial, regional or social community that will be handed down from generation to generation. North [29] believed that culture actually provides a framework for encoding and decoding all kinds of information received by the human brain.

Hofstede [13] proposed four national culture dimensions: (1) Power distance—the measurement of hierarchical distance and intensity of power between subordinates and superiors; (2) uncertainty avoidance—the measurement of members of a society avoiding uncertain situations or unpredictable events; (3) individualism versus collectivism—caring more about individual than collective benefits; and (4) masculinity versus femininity—the distinction between men's and women's roles is extremely clear or vice versa.

The risk culture of capital markets is shaped by the national culture, especially the characteristics of uncertainty avoidance, since the intrinsic nature of risk is uncertainty [14–16]. The capital market is an imperative financing channel, and the efficient allocation of resources is achieved through functions; for instance, risk transfer and risk management [30,31]. For two capital markets with different institutional qualities, such as political systems, legal environments, and many other factors, the participants have

different attitudes toward uncertainty, which will determine their judgments and behaviors with regard to risk; i.e., will form different risk cultures of the capital market.

From the perspective of the political environment, the policy uncertainty of the Chinese capital market is higher [17]. Due to different market conditions, the development of the Chinese capital market progresses by competition between regulators and traders. The opportunity emerges when regulators lag behind traders, but gradually, the contradictions are exposed and regulators start to address potential problems by developing new rules. Updated regulations lower market profits and push it to find new loopholes. A typical example is that China's stock market was on a rollercoaster ride in 2015. Regulators took numerous measures to alleviate volatility; for instance, regulations concerning short selling ranged from punishing some short sellers to forbidding malicious short selling, and then to cracking down on illegal short selling within one month [32,33]. In this way, all participants and regulators are exposed to high levels of uncertainty, which leads to the formation of a relatively strong risk culture.

From the perspective of the legal environment, the legislative process and levels of law enforcement are different among countries [34]. Due to the country's large population and complicated social environment, the legislative process of the Chinese capital market is slow and the strict enforcement of laws is difficult. In addition, some regulations have been enacted by regional rather than national regulators, which further increases uncertainty [8]. In this condition, participants have a higher expectation of risks and are more accepting of uncertainty.

Additionally, because of the incomplete formal system, the informal system has a bigger space to run in China [35]. Therefore, personal connections play a vital role in the Chinese capital market and further increase the uncertainty. Chen et al. [36] considered that political connections play an important role in economic activities, especially in emerging markets, and show that underwriter managers' political connections promote the likelihood of initial public offering (IPO) approval.

Above all, we believe the participants in the Chinese capital market have relatively low uncertainty avoidance, i.e., high acceptance and low stress in the face of uncertainty, as well as tolerance of ambiguity [27,37], which makes for a higher risk culture.

**Hypothesis 1.** *Compared with developed countries, the Chinese capital market has a higher risk culture.*

It has been found that culture influences a company's organizational design [38], compensation contracts [39], management attitudes [40], production decisions [41,42], innovation activities [18], accounting decisions and policy choices, and operating performance. Ahern et al. [43] held that the cultural gap between an acquiring company and an acquired company increases the post-acquisition integration cost and dampens the synergy effect of the acquisition. Elia et al. [18] indicated that cultural distance between partners impedes performance.

In the context of globalization, scholars have found that multinational corporations have different performances or efficiencies in different countries. Berger and Humphrey [25] argued that an overseas branch, no matter where it is established or held in stock or acquired, has a different operating efficiency compared with the company's headquarters. Furthermore, foreign financial companies have higher operating efficiency in a developing capital market due to their technological advantages [44]. However, these studies did not explore the influence of cultural differences and conflicts on company performance.

Some studies have been conducted on performance from a cultural perspective. The dominant view showed that cultural distance has a negative effect on performance because of the higher cost of coordination, management, and operation [5,6,8,9,27,45–47]. Using a sample of 10,562 overseas subsidiaries in 17 emerging market countries, Shirodkar and Konara [8] found that cultural distance resulted in poor performance. Beugelsdijk et al. [5] analyzed 156 literatures, indicating that cultural distance has a strong negative impact on the performance of subsidiaries. In contrast, some other studies have acknowledged that cultural distance has a positive or no impact on performance [48–50]. Wu and Lin found no relation between cultural distance and performance using 1596 multinational

firms. All in all, the results remain mixed, especially for emerging markets, and few studies have focused on the impact of culture on enterprises in capital markets. Although Boubakri et al. [45] argued that culture has an important impact on banks, no research has investigated how culture influences securities companies. Securities companies play an important role in the allocation of resources, involve complex financial activities, connect many participants, face diverse regulatory risks, and are more likely to be influenced by the capital market risk culture.

Due to the special market conditions in China, the establishment of IJVS is the main way for foreign companies to participate in the financial market, and those shareholders face multilayered acculturation, i.e., the cultural fit with partners should be assessed in terms of both host and home dimensions [18]. The difference of risk culture between foreign shareholders and Chinese shareholders may have two impacts on IJVS.

The first impact is on the internal management process. In multicultural organizations, communication costs are higher, making collaboration more difficult and worsening company performance [7]. IJVS usually adopt joint review and collective decision-making in management, which is quite different from the Chinese shareholders' philosophy of highlighting the main person in charge. China is a country with a larger power distance, less independence of subordinates and a more vertical organizational structure, accustomed to "leader–follower" relationships [18,27,37], while IJVS are generally composed of teams of shareholders from both sides, and their different cultural backgrounds can cause conflicts that adversely affect performance.

The second scenario is the formation of value judgments. Dixit [51] found that market uncertainty reduces a company's willingness to change, such as entering or exiting the market. Foreign shareholders may misunderstand the external factors in the Chinese market, such as policies or public opinions, while their counterparts, rooted in the Chinese capital market have developed a common cultural philosophy with the investors, companies, financial intermediaries, and regulators of the market. As mentioned in the previous section, shareholders have different attitudes toward uncertainty avoidance. This cultural difference results in foreign and Chinese shareholders adopting different value judgments when weighing the risk returns of a specific project. Notably, this mechanism is different from "information asymmetry," even though the information environment is poorer than in developed markets [8], as it is not because foreign shareholders cannot access some private information. Cultural differences allow foreign shareholders to make different value judgments based on the same information as their Chinese counterparts. In a more specific scenario, a general manager of an IJVS told us:

> Shareholders on both sides had a dispute about whether they should contract the City Development Bond (CDB), which is issued by local government financing vehicles. From the view of foreign shareholders, they cannot accept such a government bond issued by a company, and also this bond features a high risk because the company's ability to repay is highly uncertain. However, Chinese shareholders believe that such bonds are implicitly guaranteed by local government and thus have large market opportunities.

It can be seen that the risk and return evaluation system between foreign and Chinese shareholders is quite different, which results in IJVS losing some projects that are evaluated as high risk by foreign shareholders but low risk by Chinese shareholders. This cognitive difference in management and project judgment has a negative impact on the performance of IJVS. Therefore, we propose Hypothesis 2 and 3:

**Hypothesis 2.** *In China's capital market, securities companies with a stronger shareholder risk culture have better performance.*

**Hypothesis 3.** *Cultural differences in risk worsen the performance of IJVS.*

## 3. Methodology and Data

### 3.1. Sample Selection

The sample data were taken from annual reports of securities companies published by the Securities Association of China (SAC). Due to the incomplete financial information disclosure of some securities companies, we finally managed to collect information on 109 firms, including their financial statements, equities, places of registration, etc. The sample covered the period 2006 to 2016. We further identified the ownership structure by the shareholders' names and shareholding ratios disclosed in the shareholder information.

We identified 14 firms with foreign investments among all of the securities companies. However, Everbright Securities and the Bank of China International (China) Co., Ltd., cannot be counted as IJVS, as their foreign shareholders are just shell companies that have registered overseas. In addition, the foreign shareholders of Changjiang BNP Paribas Peregrine Securities and Daiwa SSC Securities withdrew their investment, and the two companies ceased operation in January 2007 and September 2014, respectively. Therefore, we excluded these four securities companies, leaving 10 IJVS in the sample.

Moreover, we collected the securities companies' negative reports from CNRDS database, such as fraud or default. We also collected economic and market development data from the National Bureau of Statistics for the headquarters location of each securities company.

### 3.2. Research Design

We employed the ordinary least squares (OLS) method to test the influence of the capital market risk culture on IJVS performance. The model is shown in Equation (1), where i represents each securities firm of the sample and t represents the year. The variables in this equation are defined in the following sections.

$$\text{Performance}_{it} = \alpha_0 + \alpha_1 \text{Culture}_{it} + \alpha_2 \text{Control}_{it} + \alpha_3 \text{Year} + \alpha_4 \text{Location} + \xi_{it} \tag{1}$$

#### 3.2.1. Measurement of Performance

The performance of a securities company can be measured in many ways. Return on assets (ROA) [19,20] is the most common indicator used to measure company performance based on accounting method. In addition, based on the economic assumptions of profit maximization and cost minimization, we can also measure the performance of a company by estimating its efficiency. By employing stochastic frontier analysis (SFA), we can calculate the gap between each observation and the efficient frontier, and the results represent the technical efficiency of the securities company [21,22].

#### Accounting Performance

This paper first employed ROA to measure the performance of a securities company. Since the revenue of China's securities companies mainly comes from traditional brokerage and securities underwriting businesses, the majority of their revenue comes from service charges and commissions. Such intermediary businesses reflect not only the performance of a securities company, but also its market recognition level.

#### Technical Efficiency

We employed SFA as an indicator of a securities company's technical efficiency. The advantage of this method is that non-efficiency terms and random disturbance terms can be distinguished.

Supposing a securities company's production possibility frontier is $y = f(z)$, its actual output is $y^{\text{real}}$ ($y^{\text{real}}$ is often lower than $y = f(z)$). When a random disturbance term v is added to the model and $y^{\text{real}}$ is kept positive, a stochastic frontier of the firm's production and operation is obtained, i.e.,

$f(z) \exp(v)$, which represents a production possibility frontier that takes into consideration random disturbance factors. Here $y^{real}$ can be represented by Equation (2):

$$y^{real} = [f(z) \exp(v)]TE \tag{2}$$

Suppose $f(z) = \alpha_0 k^{\alpha_1} l^{\alpha_2}$, where k represents capital input and l represents labor input. Take the logarithm of Equation (2), and then we can obtain the following model:

$$\ln y = \alpha_0 + \alpha_1 \ln k + \alpha_2 \ln l + v_{it} - u_{it} \tag{3}$$

In Equation (3), lny represents the logarithm of a securities company's output, which is proxy by annual total profit. lnk and lnl represent the logarithms of its capital input and labor input, respectively. We employed the total asset of securities company to measure its capital input and cost of goods sold to measure labor input. Generally, we should have measured labor input by employees' salaries, however, the annual report does not disclose the employees' salaries, therefore, in this paper, we used the cost of goods sold to approximate salaries as the labor input. $v_{it}$ is the random disturbance term and is assumed to follow a normal distribution in this paper, $u_{it}$ is a non-efficiency output term and is assumed to follow a half-normal distribution. $v_{it}$ and $u_{it}$ are assumed to be independent of each other.

Then we obtained the firm's technical efficiency by calculation, where $u_{it} = -\ln TE_{it}$, and TE represents technical efficiency. Therefore:

$$TE_{it} = \exp(-u_{it}) \tag{4}$$

### 3.2.2. Measurement of Capital Market Risk Culture

The risk culture reflects how people expect and emphasize risk [52]; as previously mentioned, some scholars have shown that the intrinsic nature of risk is uncertainty [14–16]. We applied the instrumental variable to measure the risk culture of capital markets. According to this idea and based on the research of [23,24], we applied the model described by Equation (5) to measure the role of capital market risk culture. The data we applied were from the World Bank Database.

$SPV_{it}$ refers to the general volatility of the capital market, measured by the stock price volatility of country i in time t; $GDPGrowth_{it}$ refers to the gross domestic product (GDP) growth of country i in time t; $SMR_{it}$ refers to the stock market returns; and $SMCG_{it}$ refers to the proportion of a country's stock market value in its GDP, to measure the development of its capital markets. In addition, the fixed effects on country and time dimensions are controlled in the model. We first estimated Equation (5) for the full sample of countries. This indicates the variance of the stock volatility on the capital market, which is explained by the fundamental factors (i.e., GDP growth, stock market returns, stock market contribution to GDP, etc.) of the country; while the recessive and unobserved factors are reflected in the model's residue $\xi_{it}$, and $E(\xi_{it}) = 0$.

$$SPV_{it} = \alpha_0 + \alpha_1 GDPGrowth_{it} + \alpha_2 SMR_{it} + \alpha_3 SMCG_{it} + \alpha_4 Country + \alpha_5 Year + \xi_{it} \tag{5}$$

Thus, $\xi_{it}$ reflects how much of the stock price volatility on the capital market of country i in time t is explained by recessive factors that have not been observed, such as policy shock and the rationality of participants. Its value, whether positive or negative, actually reflects how much the model fails to fit. The impact from recessive factors in $-100$ is the same as $+100$, only the meaning is different. Therefore, neither the mean nor the median reflects the country's risk culture over time; only the fluctuations of $\xi_{it}$ reflect the risk culture. Non-regular or unexplained factors are not risky if they vary gently from year-to-year, and the annual variation of residual factors describes the risk from policy change, participants' rationality, and so on. It is actually a risk that all financial participants should assume when conducting business in a certain capital market. It best describes the risk culture faced by a

securities firm in one country. Therefore, the value fluctuation brought by such recessive factors in the capital market of a certain country is defined as capital market risk culture, described by Equation (6):

$$\text{Culture}_i = \text{S.D.}(\xi_{it}) \tag{6}$$

Moreover, CultureDistance$_{ij}$ is the risk culture distance between countries i and j, defined as Equation (7):

$$\text{CultureDistance}_{ij} = \text{S.D.}(\xi_{it}) - \text{S.D.}(\xi_{jt}) \tag{7}$$

### 3.2.3. Control Variables

Referring to previous studies [53–57] and from the perspective of securities companies, performance is affected by asset, capital structure, ownership structure, reputation, and internal governance. This paper applies total asset (BrokerSize) and debt to asset ratio (BrokerLeverage) to measure the firm's size and capital structure, employing the investment proportion of its top five shareholders (Top5Ownership) to measure ownership structure. In addition, this paper applies the date of establishment to approximate the reputation of securities companies (BrokerAge).

From the perspective of the external environment, a securities company's performance is influenced by regulations, government relations, media and public opinions, and local economic conditions. This paper applies the shareholding ratio of the local government (LocalGovOwn) to measure the company's relationship with the government, employs negative news coverage frequency (BrokerScandal) to measure the potential influence of the media and public opinions, and utilizes local GDP growth (ProvincialGDP) and the marketization index [58] of each Chinese province to comprehensively measure the economic and market conditions faced by securities companies.

The details and sources of all of the above factors are shown in Table 1.

**Table 1.** Variable definitions.

| Type | Variable | Definition |
|---|---|---|
| Dependent Variable | ROA | Return on asset ratio, defined as net income divided by book value of the total asset, measured at the end of fiscal year; |
| | TE | Technical efficiency, the degree which deviates from the optimal output boundary calculated by SFA model; |
| Independent Variable | Culture | Risk culture of the capital market, measured as the standard deviation of unobserved implicit factors which may have an effect on the equity market volatility in a country's capital market in a certain year; |
| | CultureDistance | The distance of the risk cultures of capital market, defined as the difference between the two countries' risk culture of the capital market; |
| | JointVenture | An indicator variable that equals 1 if a securities company has foreign shareholders and 0 otherwise; |
| | CultureDistanceH | An indicator variable that equals 1 if the risk culture of the capital market between the foreign shareholders of a joint-venture securities company is above the median among all firms, and 0 otherwise; |

**Table 1.** *Cont.*

| Type | Variable | Definition |
|---|---|---|
| | BrokerSize | Natural logarithm of an asset measured at the end of the fiscal year; |
| | BrokerLeverage | Leverage ratio, defined as the book value of debt divided by book value of total assets measured at the end of fiscal year; |
| | BrokerAge | Firm age, measured as the number of years it has been registered; |
| | Top5Ownership | The proportion of company shares held by the top five shareholders of a securities company; |
| | LocalGovOwn | The proportion of company shares held by local government; |
| Control Variable | BrokerScandal | Natural logarithm of the media's negative news reports on a securities company since its establishment; |
| | ProvincialGDP | The GDP growth of the securities company's location in a given year; |
| | MarketIndex | Index for the market orientation of the economy [58], where a higher value indicates a more market-oriented regional economy; |
| | Governance | An indicator variable that equals 1 if the manager of IJVS has the management experience in China, others as 0; |
| | Group | An indicator variable that equals 1 if a securities company operates in the year after 2012, and 0 otherwise; |
| | GeoDistance | The distance between two countries from the CEPII database. |

## 4. Results

### 4.1. Summary Statistics

Table 2 shows the summary statistics of major variables grouped by Chinese securities companies and IJVS. It is shown from the results that the two subsamples do have significant differences in some basic characteristics. For example, the average operating performance and technical efficiency of Chinese securities companies are significantly higher compared to IJVS. In terms of Table 2, the average asset of Chinese securities companies is 16.28, while the asset of IJVS is only 14.53. The average debt to asset ratio of Chinese securities companies is about 70%, while that of IJVS is less than 35%. In addition, compared with IJVS, all Chinese securities companies were established earlier and have closer relations with the government.

**Table 2.** Summary statistics.

| Variables | Chinese | | | IJVS | | | Diff |
|---|---|---|---|---|---|---|---|
| | N | Mean | Std.Err | N | Mean | Std.Err | Mean |
| ROA | 961 | 0.030 | 0.030 | 84 | 0.013 | 0.044 | 0.017 *** |
| TE | 961 | 0.845 | 0.103 | 84 | 0.584 | 0.084 | 0.261 *** |
| BrokerSize | 961 | 16.278 | 1.444 | 84 | 14.531 | 1.304 | 1.747 *** |
| BrokerLeverage | 961 | 0.706 | 0.155 | 84 | 0.329 | 0.287 | 0.377 *** |
| BrokerAge | 961 | 13.266 | 6.722 | 84 | 6.262 | 5.156 | 7.004 *** |
| Top5Ownership | 961 | 0.699 | 0.338 | 84 | 0.882 | 0.296 | −0.183 *** |
| LocalGovOwn | 961 | 0.452 | 0.344 | 84 | 0.329 | 0.314 | 0.122 |
| BrokerScandal | 961 | 0.202 | 0.338 | 84 | 0.191 | 0.296 | 0.011 |
| ProvincialGDP | 961 | 9.842 | 0.915 | 84 | 9.863 | 0.455 | −0.021 *** |
| MarketIndex | 961 | 7.246 | 1.747 | 84 | 8.544 | 0.861 | −1.298 *** |

Note: This table reports summary statistics a by the type of securities company. *** indicates statistical significance at the 1% level. All variables are defined in the Table 1.

We analyzed the securities companies' ROA and technology efficiency by regression to test if international joint ventures have a significant impact on the performance of these firms after controlling all possible relevant factors. The regression also controlled the year fixed effect and the location fixed effect. The results are shown in Table 3. We found that the JointVenture negatively impinges on ROA

and technical efficiency at the 1% significance level in column (1) and (4), indicating that IJVS have significantly lower return on assets and technical efficiency.

**Table 3.** Joint-venture and securities companies' performance.

| Variables | (1) | (2) | (3) | (4) |
|---|---|---|---|---|
| | ROA | ROA | TE | TE |
| JointVenture | −0.013 *** | −0.000 | −0.251 *** | −0.138 *** |
| | (0.004) | (0.007) | (0.033) | (0.044) |
| BrokerSize | | 0.004 ** | | 0.051 *** |
| | | (0.002) | | (0.007) |
| BrokerLeverage | | −0.012 | | −0.003 |
| | | (0.011) | | (0.059) |
| BrokerAge | | 0.002 *** | | −0.001 |
| | | (0.001) | | (0.003) |
| BrokerAge$^2$ | | −0.000 *** | | 0.000 |
| | | (0.000) | | (0.000) |
| BrokerScandal | | 0.003 | | −0.015 * |
| | | (0.003) | | (0.009) |
| LocalGovOwn | | 0.002 | | 0.018 |
| | | (0.003) | | (0.021) |
| Top5Ownership | | −0.000 | | 0.020 |
| | | (0.005) | | (0.022) |
| ProvincialGDP | | 0.004 * | | 0.024 |
| | | (0.002) | | (0.015) |
| MarketIndex | | −0.003 ** | | −0.023 ** |
| | | (0.001) | | (0.009) |
| Year Fixed Effect | Included | Included | Included | Included |
| Location Fixed Effect | Included | Included | Included | Included |
| Constant | 0.024 *** | −0.049 * | 0.797 *** | −0.000 |
| | (0.007) | (0.029) | (0.048) | (0.153) |
| N | 1073 | 1044 | 1082 | 1045 |
| Adjusted R$^2$ | 0.291 | 0.332 | 0.440 | 0.559 |

Note: This table reports the results of regression on ROA and technical efficiency of 109 securities companies by OLS method. The dependent variable ROA, is measured as net income divided by book value of the total asset. The dependent variable TE, is measured as the degree which deviates from the optimal output boundary calculated by SFA model. The key independent variable JointVenture equals 1 if a securities firm has foreign shareholders and 0 otherwise. Definitions of other variables are provided in Table 1. Robust standard errors are displayed in parentheses. *, ** and *** indicate statistical significance at the 10%, 5%, and 1% levels, respectively.

According to the univariate test and regression results, IJVS have poorer performance than Chinese securities companies, which may result from an imbalance in the ownership structure brought by a disproportion between the equities and control power within the firm. It has been explicitly restricted by law that, at the initial stage of IJVS, the proportion of shares held by foreign shareholders should not exceed one-third, and this proportion can be increased to 49% after three years. Following this rule, in recent years, foreign shareholders in IJVS have actively increased their investment proportion to strive for greater control and profitability. This imbalance in ownership structure should be gradually improved. However, from the perspective of the time dimension and performance trends, the disadvantage of IJVS still exists, indicating that there are other implicit factors destabilizing the performance of IJVS in the Chinese capital market.

*4.2. Difference in Capital Market Risk Culture*

Table 4 lists the distribution of $\xi_{it}$ in the time dimension. In panel A, we employed a large sample for all available countries, taking 2008 as an example, and among the 81 sample countries, the minimum and maximum values of $\xi_{it}$ are −243.412 and 126.650, and the standard error is 39.455.

**Table 4.** Capital market risk culture in time dimension.

| Panel A: Distribution in time | | | | | |
|---|---|---|---|---|---|
| **Year** | **No. of Countries** | **Median** | **Min** | **Max** | **Std.Err** |
| 2008 | 81 | −2.614 | −243.412 | 126.650 | 39.455 |
| 2009 | 81 | −3.196 | −197.838 | 98.241 | 30.412 |
| 2010 | 84 | 1.283 | −141.894 | 22.279 | 19.613 |
| 2011 | 86 | −8.773 | −75.080 | 823.834 | 90.586 |
| 2012 | 86 | 5.287 | −240.689 | 49.209 | 30.171 |
| 2013 | 60 | 4.223 | −40.235 | 43.037 | 15.361 |
| 2014 | 59 | 0.634 | −48.606 | 161.041 | 26.301 |
| 2015 | 52 | −3.849 | −45.442 | 294.722 | 42.980 |
| 2016 | 52 | 1.690 | −38.346 | 26.938 | 12.607 |

| Panel B: Cross-sectional distribution | | | | | |
|---|---|---|---|---|---|
| **Year** | **China** | **US** | **Switzerland** | **France** | **UK** | **Germany** |
| 2008 | −132.68 | 81.86 | 47.25 | 29.73 | −11.52 | 126.66 |
| 2009 | 36.08 | 41.42 | −2.12 | −7.20 | 38.48 | 26.23 |
| 2010 | −17.89 | 21.94 | 9.90 | 3.72 | 0.39 | 0.83 |
| 2011 | −75.08 | 33.29 | −9.05 | −3.76 | −16.05 | −5.23 |
| 2012 | −90.41 | −9.60 | −9.86 | 3.53 | −11.30 | −9.86 |
| 2013 | −30.36 | −40.24 | −12.36 | −11.42 | | −31.24 |
| 2014 | 3.02 | −47.27 | −16.19 | −14.61 | | −41.12 |
| 2015 | 294.72 | −45.44 | −6.63 | | | −27.93 |
| 2016 | 12.60 | −35.97 | −0.95 | | | −38.35 |

| Panel C: Summary statistics | | | | |
|---|---|---|---|---|
| **Country** | **Min** | **Max** | **Culture$_i$ = S.D. ($\xi_{it}$)** | **CultureDistance$_{ij}$** |
| China | −132.68 | 294.72 | 122.98 | 0 |
| US | −47.27 | 81.86 | 44.35 | 78.63 |
| Switzerland | −16.19 | 47.25 | 18.70 | 104.28 |
| France | −14.61 | 29.73 | 14.83 | 108.15 |
| UK | −16.05 | 38.48 | 22.35 | 100.63 |
| Germany | −41.12 | 126.66 | 52.17 | 70.81 |

Note: This table reports the distributions of residual $\xi_{it}$. Panel A reports distribution of residual in time dimension. Panel B reports distribution of residual in cross-sectional dimension. Panel C reports capital market risk culture and cultural distance between investment origination country and China. Definitions of other variables are provided in Table 1.

Panels B and C report countries that make investments in Chinese securities firms; that is, the countries we investigated in the following part. According to the shareholders' home country, panel B lists the distribution of $\xi_{it}$ in China, the United States, Switzerland, France, Britain, and Germany at different times, which shows a huge difference.

We calculated the capital market risk culture in panel C. The value of China's risk culture is 122.98; in the same way, the United States is 44.35, Switzerland is 18.70, France is 14.83, Britain is 22.35, and Germany is 5.17. The volatility of residuals shows that China has the strongest risk culture among all of these countries. This conclusion indicates that Chinese capital market participants have higher uncertainty acceptability or risk tolerance, which supports H1.

Column 4 of panel C calculates the gap between other countries and China in the value of risk culture, to show how much risk culture difference foreign shareholders experience in China's capital market. Judging from the values of CultureDistance$_{ij}$, the difference in risk culture between China and the United States and that between China and Germany are relatively small, while that between China and Switzerland, France, and Britain is larger. As a result, when foreign shareholders of IJVS come from the United States or Germany, they will face smaller risk culture differences in China's capital market, but if the shareholders are from Britain, France, or Switzerland, the cultural impact will be greater.

### 4.3. Influences of Capital Market Risk Culture on the Performance of Securities Companies

This paper examined the impact of risk culture on securities companies' performance in the following analysis. We approximated the company's risk culture by the nature of its shareholders. For Chinese securities firms, a company is effectively controlled by its Chinese parent company. Since the investment judgments and behaviors of these shareholders are formed in China's capital market, we believe their risk culture is the Chinese risk culture (i.e., Culture = 122.98). For IJVS, foreign shareholders actually control the company and the form of risk culture is rooted in the capital market of the home country. Therefore, this paper assigned the value of IJVS risk culture by the risk culture of the foreign shareholders' home country.

Table 5 shows that the higher the value of risk culture, the higher the return on assets (ROA) and technical efficiency of the securities firm, which supports H2. BrokerSize has a significant positive impact on firm performance, and this finding is consistent with previous research [55]. Additionally, we found that market index has a significantly negative impact on firm performance. This result is different from some previous research [57,59]. One possible explanation is that most IJVS are located in cities with good market conditions, for example, Shanghai and Shenzhen. Another possible reason is that the securities companies are supported by local governments in cities with poor market conditions, which may promote the efficiency of these companies. Moreover, the relation between ROA and leverage is significantly negative, consistent with [56], and the relation between firm's age and performance is also consistent with previous research [60].

**Table 5.** Risk culture of capital market and securities companies' performance.

| Variables | (1) ROA | (2) TE |
|---|---|---|
| Culture | 0.002 *** | 0.082 *** |
| | (0.001) | (0.029) |
| BrokerSize | 0.005 *** | 0.050 *** |
| | (0.001) | (0.008) |
| BrokerLeverage | −0.018 ** | −0.027 |
| | (0.009) | (0.051) |
| BrokerAge | 0.002 *** | 0.000 |
| | (0.001) | (0.003) |
| BrokerAge$^2$ | −0.000 *** | 0.000 |
| | (0.000) | (0.000) |
| BrokerScandal | 0.001 | −0.013 |
| | (0.002) | (0.009) |
| LocalGovOwn | 0.004 | 0.031 |
| | (0.003) | (0.020) |
| Top5Ownership | −0.002 | 0.002 |
| | (0.003) | (0.018) |
| ProvincialGDP | 0.003 * | 0.025 * |
| | (0.002) | (0.015) |
| MarketIndex | −0.002 ** | −0.023 *** |
| | (0.001) | (0.008) |
| Year Fixed Effect | Included | Included |
| Location Fixed Effect | Included | Included |
| Constant | −0.061 *** | −0.383 *** |
| | (0.018) | (0.139) |
| N | 1026 | 1027 |
| Adjusted R$^2$ | 0.360 | 0.551 |

Note: This table reports OLS regression estimates of risk culture on ROA and technical efficiency of 109 securities companies. The dependent variable ROA, is measured as net income divided by book value of the total asset. The dependent variable TE, is measured as the degree which deviates from the optimal output boundary calculated by SFA model. The key independent variable Culture, measured as the standard deviation of unobserved implicit factors which may have an effect on the equity market volatility in a country's capital market in a certain year. Definitions of other variables are provided in Table 1. Robust standard errors clustered by firm are displayed in parentheses. *, ** and *** indicate statistical significance at the 10%, 5%, and 1% levels, respectively.

### 4.4. Influences of Cultural Distance on the Performance of Securities Companies

To further test the business disadvantage of risk culture differences, we took IJVS as a sample to estimate their performance related to the risk culture distance between the Chinese capital market and foreign shareholders.

As shown in Table 4, IJVS with foreign shareholders from the United States or Germany (including China International Capital Corporation Limited, Citi Orient Securities Co., Ltd., Goldman Sachs Gao Hua Securities Co., Ltd., J.P. Morgan First Capital Securities Co., Ltd., Morgan Stanley Huaxin Securities Co., Ltd., and Zhong De Securities Co., Ltd.) face smaller risk culture differences compared to the Chinese capital market, with the difference value staying at 70.81 and 78.68, respectively. However, those with foreign shareholders from Britain, France, or Switzerland (including Hua Ying Securities Co., Ltd., CEFC Shanghai Securities Co., Ltd., UBS Securities Co., Ltd. and Credit Suisse Founder Securities Limited) face larger risk culture differences compared to the Chinese capital market, with the difference value staying at 108.15, 104.28, and 100.63, respectively.

After adding control variables and the risk culture distance of securities companies, the regression results based on 10 IJVS, 76 "sample-year" observations were formed, as shown in Table 6. This analysis also controlled the governance imbalance factor, which features a mismatch between shareholding and control that may affect the business performance of an IJVS. The result shows that the coefficient of CultureDistance is significantly negative, indicating that the larger the gap in risk culture, the lower the return on assets and technical efficiency of IJVS, which supports H3.

**Table 6.** Culture distance and securities companies' performance.

| Variables | (1) ROA | (2) TE |
|---|---|---|
| CultureDistance | −0.033 ** | −0.082 *** |
| | (0.017) | (0.039) |
| Governance | 0.001 | −0.003 |
| | (0.001) | (0.005) |
| BrokerSize | 0.014 | 0.050 |
| | (0.013) | (0.050) |
| BrokerLeverage | 0.023 | −0.133 |
| | (0.029) | (0.092) |
| BrokerAge | 0.006 | −0.003 |
| | (0.003) | (0.011) |
| BrokerAge$^2$ | −0.001 ** | −0.000 |
| | (0.000) | (0.000) |
| BrokerScandal | 0.043 | 0.007 |
| | (0.031) | (0.047) |
| LocalGovOwn | 0.063 * | −0.001 |
| | (0.033) | (0.047) |
| Top5Ownership | 0.023 * | −0.002 |
| | (0.010) | (0.019) |
| ProvincialGDP | −0.664 | −1.326 |
| | (0.378) | (0.807) |
| MarketIndex | 0.013 | −0.074 * |
| | (0.035) | (0.036) |
| Year Fixed Effect | Included | Included |
| Location Fixed Effect | Included | Included |
| Constant | 6.463 | 14.193 |
| | (3.562) | (7.780) |
| N | 76 | 76 |
| Adjusted R$^2$ | 0.499 | 0.460 |

Note: This table reports OLS regression estimates of cultural distance on ROA and technical efficiency of 10 IJVS. The dependent variable ROA, is measured as net income divided by book value of the total asset. The dependent variable TE, is measured as the degree which deviates from the optimal output boundary calculated by SFA model. The key independent variable CultureDistance, measured as the difference between the two countries' risk culture. Definitions of other variables are provided in Table 1. Robust standard errors clustered by firm are displayed in parentheses. *, ** and *** indicate statistical significance at the 10%, 5%, and 1% levels, respectively.

### 4.5. Cross-Sectional Test

If the risk culture difference leads IJVS to different judgments of risk vs. return on specific projects, and therefore brings disadvantages, then their performance may even worsen when they expand the business scope.

The test in this part aimed to discuss if the impact of a larger risk culture distance varies when IJVS have more diversified business. For securities companies, licenses are a prerequisite for business, and scholars found that some important resources such as licenses are preferentially accessible to state-owned enterprises in many emerging markets [8]. Before 2012, most IJVS only had investment banking licenses, which restricted their business scope. From 2012, the China Securities Regulatory Commission gradually broadened the business scope of IJVS and allowed them to obtain all financial business licenses, such as trust business, futures business, or insurance business. Following the policy changes, this paper set 2012 as the boundary and divided the samples into two groups, before 2012 (Group = 0) and after 2012 (Group = 1). We added an interaction term in the new regression, obtained by multiplying the culture distance term (CultureDistance) with the group term (Group). The result, shown in Table 7, shows the performance of IJVS improving after 2012, but the interaction term is negative. This means that the negative impact of risk culture distance on performance gets stronger after IJVS have access to more diversified businesses.

**Table 7.** A cross-sectional test on the culture distance and performance of securities companies.

| Variables | (1) ROA | (2) TE |
|---|---|---|
| CultureDistance | −0.015 | −0.021 |
| | (0.027) | (0.063) |
| Group | 0.385 ** | 0.790 ** |
| | (0.141) | (0.299) |
| CultureDistance×Group | −0.025 * | −0.083 ** |
| | (0.014) | (0.041) |
| Governance | 0.002 | −0.000 |
| | (0.003) | (0.001) |
| BrokerSize | 0.016 | 0.052 |
| | (0.013) | (0.049) |
| BrokerLeverage | 0.023 | −0.131 |
| | (0.032) | (0.094) |
| BrokerAge | 0.006 | −0.003 |
| | (0.004) | (0.009) |
| BrokerAge$^2$ | −0.001 ** | −0.000 |
| | (0.000) | (0.000) |
| BrokerScandal | 0.037 | −0.015 |
| | (0.032) | (0.049) |
| LocalGovOwn | 0.057 | −0.028 |
| | (0.034) | (0.060) |
| Top5Ownership | 0.024 ** | 0.011 |
| | (0.011) | (0.021) |
| ProvincialGDP | −0.667 * | −1.309 * |
| | (0.356) | (0.657) |
| MarketIndex | 0.013 | −0.072 * |
| | (0.037) | (0.039) |
| Year Fixed Effect | Included | Included |
| Location Fixed Effect | Included | Included |
| Constant | 6.093 * | 13.199 * |
| | (3.154) | (5.917) |
| N | 76 | 76 |
| Adjusted R$^2$ | 0.509 | 0.503 |

Note: This table reports OLS regression estimates of cultural distance on ROA and technical efficiency of 10 IJVS, and the mediating effect of business scope. The dependent variable ROA, is measured as net income divided by book value of the total asset. The dependent variable TE, is measured as the degree which deviates from the optimal output boundary calculated by SFA model. The key independent variable CultureDistance, measured as the difference between the two countries' risk culture of capital market. Group equals 1 when a securities company operates in the year after 2012, and 0 otherwise. Definitions of other variables are provided in Table 1. Robust standard errors clustered by firm are displayed in parentheses. * and ** indicate statistical significance at the 10%, 5%, and 1% levels, respectively.

*4.6. Robustness Tests*

4.6.1. Alternative Measurement of Capital Market Risk Culture

We applied the alternative measurement of capital market risk culture to test if our conclusions are robust. We used a dummy variable (CultureDistanceH) to measure the risk culture distance in our regression. The value was set as 1 when the risk culture difference of foreign shareholders to Chinese shareholders was bigger than the sample median, otherwise 0. After adding control variables, the regression results based on 10 IJVS, 76 "securities-year" observations were formed, as shown in Table 8.

**Table 8.** A robustness test on the culture distance and performance of IJVS.

| Variables | (1) ROA | (2) TE |
|---|---|---|
| CultureDistanceH | −0.029 ** | −0.070 ** |
| | (0.010) | (0.029) |
| Governance | 0.000 | −0.001 |
| | (0.001) | (0.005) |
| BrokerSize | 0.014 | 0.050 |
| | (0.013) | (0.049) |
| BrokerLeverage | 0.022 | −0.132 |
| | (0.029) | (0.092) |
| BrokerAge | 0.006 | −0.003 |
| | (0.003) | (0.011) |
| BrokerAge$^2$ | −0.001 ** | −0.000 |
| | (0.000) | (0.000) |
| BrokerScandal | 0.044 | 0.005 |
| | (0.030) | (0.046) |
| LocalGovOwn | 0.063 * | −0.002 |
| | (0.033) | (0.047) |
| Top5Ownership | 0.022 * | 0.000 |
| | (0.010) | (0.021) |
| ProvincialGDP | −0.665 | −1.322 |
| | (0.374) | (0.801) |
| MarketIndex | 0.013 | −0.074 * |
| | (0.036) | (0.037) |
| Year Fixed Effect | Included | Included |
| Location Fixed Effect | Included | Included |
| Constant | 6.441 * | 14.062 |
| | (3.511) | (7.694) |
| N | 76 | 76 |
| Adjusted R$^2$ | 0.501 | 0.463 |

Note: This table reports OLS regression estimates of cultural distance on ROA and technical efficiency of 10 IJVS, and the mediating effect of business scope. The dependent variable ROA, is measured as net income divided by book value of the total asset. The dependent variable TE, is measured as the degree which deviates from the optimal output boundary calculated by SFA model. The key independent variable CultureDistanceH, equals 1 if the risk culture of capital market between the foreign shareholders of a joint-venture securities company is above the median among all firms, and 0 otherwise. Definitions of other variables are provided in Table 1. Robust standard errors clustered by firm are displayed in parentheses. * and ** indicate statistical significance at the 10%, 5%, and 1% levels, respectively.

The results for ROA in column 1 show that the coefficient of CultureDistanceH is negative and significant at the 5% level, suggesting that IJVS with a larger difference in risk culture from China have 2.9% lower ROA on average than those with a smaller difference. The regression results of technical efficiency in column 2 show that the coefficient of CultureDistanceH is negative and significant at the 5% level, suggesting that IJVS with a larger difference of risk culture from China have 0.07 lower TE on average than those with a smaller difference.

The above results further prove our hypothesis that the risk culture distance between foreign shareholders and the Chinese capital market has a significant impact on the performance of IJVS. The larger the cultural gap, the greater the impact on performance.

### 4.6.2. Test Based on Matched Samples

The analysis based on the full sample suggests that the risk culture has an impact on the performance of securities companies. However, a potential problem is that securities companies of different sizes may be various in their business structure and strategic goals. Taking IJVS as an example, their foreign shareholders are mostly well-funded and, internationally competitive investment banks are looking to become market leaders when entering the Chinese market. Such a strategic goal is largely different from small- or medium-sized local securities companies. Therefore, it is not persuasive enough to directly compare the performance between IJVS and local small or medium companies in the analysis.

To solve this problem, we matched IJVS with Chinese securities companies according to the date of establishment, place of headquarters, and asset, selecting 20 comparable Chinese securities companies to match the 10 IJVS, generating almost 290 "firm-year" observations from 30 securities companies. We analyzed the ROA and TE of these firms by OLS regression, and the results are shown in Table 9. The results suggest that in the matched sample, the risk culture still has a significant positive influence on ROA and technological efficiency, i.e., securities companies with a stronger risk culture have better performance.

**Table 9.** Matched samples of the risk culture and performance of securities companies.

| Variables | (1) ROA | (2) TE |
|---|---|---|
| Culture | 0.002 ** | 0.108 ** |
| | (0.001) | (0.041) |
| BrokerSize | 0.002 | 0.040 ** |
| | (0.002) | (0.019) |
| BrokerLeverage | −0.004 | −0.086 |
| | (0.013) | (0.085) |
| BrokerAge | 0.004 ** | 0.007 |
| | (0.002) | (0.008) |
| BrokerAge$^2$ | −0.000 *** | −0.000 |
| | (0.000) | (0.000) |
| BrokerScandal | 0.006 | 0.010 |
| | (0.006) | (0.024) |
| LocalGovOwn | 0.001 | 0.040 |
| | (0.008) | (0.040) |
| Top5Ownership | −0.016 * | −0.022 |
| | (0.008) | (0.041) |
| ProvincialGDP | 0.010 ** | 0.055 |
| | (0.004) | (0.038) |
| MarketIndex | −0.002 | −0.019 |
| | (0.008) | (0.057) |
| Year Fixed Effect | Included | Included |
| Location Fixed Effect | Included | Included |
| Constant | −0.095 | −0.628 |
| | (0.067) | (0.425) |
| N | 286 | 287 |
| Adjusted R$^2$ | 0.245 | 0.634 |

Note: This table reports OLS regression estimates the impact of risk culture on company performance of 10 IJVS and 20 comparable securities companies. The dependent variable ROA, is measured as net income divided by book value of the total asset. The dependent variable TE, is measured as the degree which deviates from the optimal output boundary calculated by SFA model. The key independent variable Culture, measured as the standard deviation of unobserved implicit factors which may have an effect on the equity market volatility in a country's capital market in a certain year. Definitions of other variables are provided in Table 1. Robust standard errors clustered by firm are displayed in parentheses. *, ** and *** indicate statistical significance at the 10%, 5%, and 1% levels, respectively.

### 4.6.3. Test Using Instrumental Variables

Endogeneity is a common challenge seen in finance and economics literature. This paper tried to reduce this concern by including as many control variables as possible, introducing fixed effects to deal with the situation in which some other variable affects culture and performance simultaneously [61], and employed the propensity score matching (PSM) method.

However, it is worth noting that we can hardly interpret that the performance of IJVS causes cultural distance between two countries, while the interpretation reasonably holds in another direction. By no means do we aim to prove that cultural factors are the only cause of inferior performance of IJVS. Instead, we want to provide evidence that it is one of the main drivers.

To mitigate residual endogeneity, we employed an instrumental variable approach [61]. The instrument we used was geographic distance. While geographic distance is a key driver of cultural distance [62], no direct evidence shows that it influences firm performance. We obtained the geographic distance data between two countries (GeoDistance) from the CEPII database. Then, we re-estimated the regression to test the relation between cultural distance and the performance of IJVS. The results are shown in Table 10, which indicates that the results are consistent with the analysis above.

**Table 10.** Geographic distance and performance of securities companies.

| Variables | (1) ROA | (2) TE |
|---|---|---|
| GeoDistance | −0.001 ** | −0.002 ** |
| | (0.000) | (0.001) |
| Governance | 0.000 | −0.001 |
| | (0.001) | (0.002) |
| BrokerSize | 0.016 | 0.052 ** |
| | (0.012) | (0.025) |
| BrokerLeverage | 0.026 | −0.122 * |
| | (0.033) | (0.068) |
| BrokerAge | 0.005 | −0.005 |
| | (0.003) | (0.007) |
| BrokerAge$^2$ | −0.001 *** | −0.000 |
| | (0.000) | (0.000) |
| BrokerScandal | 0.047 *** | 0.004 |
| | (0.015) | (0.031) |
| LocalGovOwn | 0.063 ** | −0.012 |
| | (0.026) | (0.054) |
| Top5Ownership | 0.020 | −0.029 |
| | (0.020) | (0.041) |
| ProvincialGDP | −0.612 *** | −1.054 ** |
| | (0.219) | (0.454) |
| MarketIndex | 0.004 | −0.102 * |
| | (0.025) | (0.052) |
| Year Fixed Effect | Included | Included |
| Location Fixed Effect | Included | Included |
| Constant | 5.857*** | 11.349** |
| | (2.079) | (4.306) |
| N | 76 | 76 |
| Adjusted R$^2$ | 0.486 | 0.439 |

Note: This table reports IV regression estimates the impact of risk culture on company performance of 10 IJVS securities companies. The dependent variable ROA, is measured as net income divided by book value of the total asset. The dependent variable TE, is measured as the degree which deviates from the optimal output boundary calculated by SFA model. The instrumental variable GeoDistance, measured as the distance between two countries from the CEPII database. Definitions of other variables are provided in Table 1. Robust standard errors clustered by firm are displayed in parentheses. *, ** and *** indicate statistical significance at the 10%, 5%, and 1% levels, respectively.

Moreover, there still exist some endogeneity problems; for example, chief executive officer (CEO) characteristics including executive compensation and CEO delta [63,64]. In addition, the limitation of our sample is that the vast majority are non-listed companies; therefore, all data needed to be collected manually, and only a few companies had disclosed data related to their CEOs. Although this is a limitation of this paper, it also provides a new direction for solving the endogeneity problem.

## 5. Conclusions

This paper studied the influence of capital market risk culture and cultural distance on the performance of IJVS. Based on securities companies' data from 2006 to 2016 published by the Securities Association of China (SAC), we first found that the risk culture of the Chinese capital market is stronger than that in developed countries. Second, we found that in the Chinese capital market, the securities companies with a stronger risk culture can achieve better performance. Third, we found that the larger the risk culture distance, the greater the negative impact on the performance of IJVS. Moreover, in a cross-sectional test, we found that the negative impact of risk culture distance on performance is greater when IJVS expand their business territory.

Considering that our database covered many small Chinese securities companies that have great differences from IJVS, which may cause bias in our conclusion, we further selected 20 comparable Chinese securities companies to pair with the 10 IJVS according to the date of establishment, place of headquarters, and asset. The analysis based on a matched sample found that IJVS still have significantly poorer performance and technical efficiency compared with Chinese securities companies. The analysis that employed geographical distance as an instrumental variable still proved that cultural distance impedes IJVS performance.

This paper provides new evidence for the factors that may influence the performance of financial institutions in emerging capital markets. It was found that risk culture plays a vital role in the development of international financial institutions and FDI. These findings can help managers of multinational companies to pay more attention to the role of culture in transnational business. Some measures need to be implemented at various levels to promote sustainable FDI (e.g., learning the cultures of other countries, holding regular consultative meetings between both sides, offering joint staff training, etc.), and thus may help more enterprises to achieve sustainable competitiveness in transnational business.

There are also some limitations of our research. First, the small sample size is not ideal and does not cover the most recent years. This is because our data were manually collected, which was a labor-intensive process, and another reason is that some IJVS went bankrupt or left the Chinese market after 2016. Second, we applied SFA to calculate firm efficiency and ROA for company performance, and the conclusion could be more robust if we introduced other indicators, for example, economic value added (EVA) and alpha for firm performance [65,66]. Finally, there still remain other factors that may influence the sustainability of companies. Li et al. [66] emphasized the importance of corporate social responsibility in performance; we may manually collect data in the future to verify whether it also works for the sustainability of securities companies.

**Author Contributions:** Conceptualization, X.M. and D.W.; data curation, X.M.; formal analysis, D.W. and H.G.; investigation, Z.T.; methodology, D.W.; project administration, X.M.; resources, Z.T.; software, D.W.; supervision, X.M.; validation, X.M. and Z.T.; visualization, X.M. and D.W.; writing—original draft preparation, D.W. and H.G.; writing—review and editing, X.M. and Z.T. All authors have read and agreed to the published version of the manuscript.

**Funding:** This research received no external funding.

**Conflicts of Interest:** The authors declare no conflict of interest.

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
