# Peer review of "The Influence of Risk Culture on the Performance of International Joint-Venture Securities"

_sustainability, doi:10.3390/su12072603_

Round 1
Reviewer 1 Report
This paper examines the influence of the capital market risk culture and the culture distance on the performance of IJVS, based on the securities’ data from 2006 to 2016 published by the Securities Association of China (SAC).
The authors should have supported their work in a theory that would help them to support their hypotheses and the results achieved.
They should also justify the analysis period and did not do so. Why do they finish the analysis in 2016? They should complement the study with the analysis of more recent years.
The authors could support their literature review with more current references. They only present one reference of 2019, one of 2018 and three of 2017.
Moreover, they did not adequately support their variables with previous studies, nor the expected signal from them. In this sequence, the results could corroborate conclusions from previous studies to make them more robust and the authors did not make such a comparison. The authors could engage more in the robustness of their data.
In the conclusions of the study, they should have stated the limitations of the work and they did not do it either.
Reviewer 2 Report
Review of “The Influence of Risk Culture on the Performance of International Joint-venture Securities”
This paper tests the following research hypotheses:
(H1) According to the authors, the risk aversion coefficient in Chinese investors' utility functions is lower than the same parameter in foreign investors' utility functions. The authors interpret this higher risk tolerance of Chinese investors as a higher risk culture for Chinese investors that for foreign investors (Hypothesis 1). The authors explain this by referring to the higher policy uncertainty and imperfect legal system in China.
(H2) The authors also study whether the performances of international joint venture companies benefit from the higher risk culture of Chinese investors (Hypothesis 2). According to the second hypothesis joint venture companies benefit from the higher risk culture of Chinese investors.
(H3) The authors also investigate if the cultural differences between foreign investors and Chinese investors are great, then, due to those differences, profitable projects in China are avoided, which result lower firm performances (Hypothesis 3). Thus, according to this hypothesis, higher cultural differences imply lower firm performance.
The dataset that is used to these hypotheses are for 109 Chinese financial companies for the period of 2006 to 2016. From those 109 financial firms the authors identify 10 financial firms with foreign investment. Data on those 10 financial firms are used to test the hypotheses.
Question: Is the sample for which the econometric models are estimated includes the 109 firms or only the 10 firms? (Table 1 motivates this question in which a dummy variable is presented that indicated foreign investment.) Please clarify this in the paper.
The econometric model is a panel data model in which the dependent variable is firm performance and the explanatory variables are capital market risk culture of shareholders and several control variables.
Minor comment: OLS is not a model, but it is an estimation method. The model according to Equation (1) is a linear panel data model without unobserved heterogeneity and time effects.
Question: In Equation 1, index i is i=1,...,10 for each of the financial firms of the sample? Please clarify this in the paper.
Firm performance is measured by using (a) ROA and (b) stochastic frontier analysis. The latter is a measurement of technical efficiency for each firm. The level of risk culture is measured by using the residuals of a linear panel data model in which the dependent variable is the stock market volatility of country i and the explanatory variables are different country i-specific macroeconomic variables and capital market variables.
Question: Are the countries considered in this second panel data model the same countries from which the foreign investments to the Chinese financial firms go? Please clarify this in the paper.
Question: According to Equation (6), the risk culture level variable is defined as the standard deviation of the error term in the second panel data model (Equation (5)). This implies that the estimation method that is used for Equation (5) uses heteroscedastic (country-dependent and time-dependent) variance for the error term. How those standard deviations are estimated?
I have doubts that the standard deviation of the error of Equation (5) is an appropriate measurement of country risk culture. Please convince better about this in the paper.
In Table 1, the authors may explicitly indicate which of the variables are dependent variable and which are control variables.
Question: The authors use the OLS estimator for the panel data model of Equation (1) which assumes that all explanatory variables are exogenous. Are all the control variables exogenous? Did the authors consider the use of the GMM estimator which is robust to endogenous explanatory variables?
Question: Please clarify the sample sizes in Table 2 for Chinese and IJVS according to the sample period and the number of firms of the sample.
Question: The panel data results that are reported in Table 3 are not specified as an econometric model before in the paper. Does that table report result for testing the first hypothesis? Please clarify.
I am having methodological doubts about the results that are reported in Section 4.2. Please see my previous comment about the estimation of the panel data model of Equation (1).
In Section 4.3, the OLS estimate of Equation (1) are reported. How do the authors estimate the standard errors of parameters? Do they use the HAC estimator or another robust estimator?
Please clarify the number of observations in Tables 5 and 6. Relate them with the sample period and the number of firms in the sample.
Reviewer 3 Report
attached

Round 2
Reviewer 1 Report
The authors have made significant efforts to address the comments, and I believe that the paper has benefited and improved significantly as a result of these changes. Although the article is not brilliant, it meets the conditions to be published.
Author Response
Dear reviewer,
Thank you very much for your interest in this paper as well as your extremely useful comments in every round of revision. We have learned a lot from your professional suggestions. We will be happy to address any remaining problems at any time.
Reviewer 2 Report
Review of "The Influence of Risk Culture on the Performance of 2 International Joint-Venture Securities"
Summary:
The authors responded well to my previous comments.
Minor comments:
The abbreviation IJVS appears ones before its definition. Please define IJVS at its first appearance.
Line 43: "However, the performance of these IJVS are not well-performed, and a number of IJVS are 43 even facing years of losses." ... performance ... well-performed; please improve this sentence.
Line 67: "The establishment of China’s financial market was rather later," The recent establishment of China’s financial market....
As an additional last paragraph of the introduction section, please include the structure of the remainder of the paper.
After Eq. (1) the authors may include "The variables in this equation are defined in the following sections."
Indices of some variables are missing from Eq. (3).
Line 320: Control variables instead of "Controlled Variables"
Based on Eq. (3), I believe that the correct formula for f(z) is:
f(x)=alpha_{0}k^{alpha_{1}}l^{alpha_{2}}
Define the TE abbreviation when it first appears in the paper.
In Table 2 there are only ***, thus only explain *** in the table notes for that table.
The titles of Sections 4.3 and 4.4 are identical.
Reviewer 3 Report
Significantly improved. Do a final proofread to correct some awkward sentences and make logic flow more smoothly.
Round 3
Reviewer 3 Report
good job. congrats!
Author Response
Dear reviewer,
Thank you very much for your interest in this paper as well as your extremely useful and detailed comments in this round of revision. We have learned a lot from your professional suggestions. We will be happy to address any remaining problems at any time.